# Tie-2 regulates endothelial morphological responses to shear stress by FOXO1-triggered autophagy

**Mana Ghanbarpour Houshangi**, **Keisuke Shirakura\***, **Dietmar Vestweber\***

Department of Vascular Cell Biology, Max Planck Institute for Molecular Biomedicine, Münster, Germany

☯ These authors contributed equally to the present study.
* vestweb@mpi-muenster.mpg.de (DV); keisuke.shirakura@mpi-muenster.mpg.de (KS)

## Abstract

### Introduction

Endothelial cells respond to flow-induced shear stress by morphological changes, a process which is important for vascular development and physiology. High laminar shear stress activates Tie-2 which supports endothelial junction integrity and protects against vascular leaks and the generation of atherosclerotic plaques.

### Methods

We have examined the role of Tie-2 and FOXO1 in controlling vascular endothelial cell morphology under physiological shear stress. To address this, we exposed human umbilical vein endothelial cells (HUVECs) transfected with siRNA to 15 dyn/cm$^2$ of shear stress for 24 hours. The resulting cells were analyzed by immunofluorescence staining.

### Results

We found that shear stress-induced activation of Tie-2 is required for endothelial cell alignment and elongation in the direction of flow. Mechanistically, we found that FOXO1 is an essential target downstream of Tie-2, which becomes translocated from the nucleus into the cytosol. There, FOXO1 stimulates the formation of autophagosomes, and both FOXO1 and autophagy stimulation are needed for Tie-2-dependent cell alignment.

### Conclusion

In conclusion, laminar fluid shear stress stimulates a novel Tie-2-FOXO1-autophagy signaling axis which is required for endothelial cell alignment. This represents a new mechanism by which Tie-2 contributes to vascular protection under laminar shear stress.

**Data availability statement:** All relevant data are within the manuscript and its Supporting Information files.

**Funding:** This work was supported by the Max Planck Society (vasb A1) to DV. The funder had no role in study design, data collection and analysis, decision to publish, or preparation of the manuscript.

**Competing interests:** The authors have declared that no competing interests exist.

## Introduction

Fluid shear stress, a frictional force generated by blood flow, plays a crucial role in vascular physiology and pathology [1–3]. Mechano-receptors in endothelial cells sense shear stress and control cell morphology [4–15]. Laminar shear stress induces cell alignment and elongation in the direction of flow, which depends on cytoskeleton remodeling [16–19]. These morphological changes help to reduce the force gradient that shear stress exerts on the cellular surface [20]. As a result, cellular sensitivity to shear stress is reduced, leading to altered endothelial responses to blood flow.

The tyrosine kinase receptor Tie-2 is a critical regulator of vascular remodeling and barrier functions [21–24]. Tie-2 acts as a receptor for angiopoietin (Angpt) ligands, and its signaling activity is modulated by the orphan receptor Tie-1 and vascular endothelial protein tyrosine phosphatase (VE-PTP) [21,23]. In addition to angiopoietins, shear stress is another prominent stimulus that influences Tie-2. Physiological flow was reported to promote Tie-2 activity by transcriptional upregulation and phosphorylation [25–32], which protect against vascular leaks and the formation of atherosclerotic plaques in mice [31,32].

Forkhead box O 1 (FOXO1) is a downstream effector of Tie-2 signaling. FOXO1 acts both as a nuclear transcription factor [33–37] and as a cytoplasmic promoter of autophagy in a transcription-independent way [38–41]. Activated Tie-2 leads to FOXO1 translocation from nuclei to cytoplasm, which inactivates its transcriptional activity [42]. However, a link of this Tie-2-FOXO1 axis to cell morphological responses triggered by shear stress remains unexplored.

In this study, we investigated whether and if so, how Tie-2 plays a role in controlling cell morphology in response to shear stress. We found that Tie-2 was essential for adapting cell morphology to shear stress. Mechanistically, Tie-2 activation led to FOXO1 translocation from nuclei to cytoplasm and subsequently promoted autophagosome formation, which was required for cell morphological responses. Thus, our findings provide evidence that Tie-2 controls cell alignment and elongation and does so through autophagy stimulated by cytoplasmic FOXO1.

## Materials and methods

### Ethics statement

The experiments using HUVECs were approved by Ethics Committee of Münster University Clinic (Approval 2009–537-f-S) and according to the principles outlined in the Declaration of Helshinki. Informed written consents were obtained from patients before donating the umbilical cords. The author had no contact with the patients and had not access to any identifying information.

### Cell culture

Human umbilical vein endothelial cells (HUVEC) were kindly provided from Prof. Hans-J. Schnittler (Münster University, Germany). HUVECs were isolated as described [43], cultured in EBM-2 medium supplemented with SingleQuots (Lonza) at 37°C and 5% $CO_2$ and used for experiments between passages 3 and 6.

## Antibodies

The following antibodies were used for immunofluorescence and immunoblotting: Rabbit monoclonal anti-human **Atg5** (clone D5F5U #12994, Cell Signaling, 1:1000 for immunoblotting); rabbit monoclonal anti-human **FOXO1** (clone C29H4 #2880, Cell Signaling, 1:1000 for immunoblotting and 1:200 for immunofluorescence staining); rabbit monoclonal anti-human **LC3B** (clone D11 #3868, Cell Signaling, 1:1000 for immunoblotting and 1:200 for immunofluorescence staining); mouse monoclonal anti-human **α-tubulin** (clone B-5-1-2 #T6074, Sigma, 0.5 µg/ml for immunoblotting); mouse monoclonal anti-human **VE-cadherin** (clone F-8 #sc-9989, Santa Cruz Biotechnology, 1:100 for immunofluorescence staining). Secondary antibodies conjugated to Alexa Fluor 488, Alexa Flour 568 (1:500) and donkey anti mouse or rabbit horseradish peroxidase (1:10000) were from Invitrogen and Jackson ImmunoResearch respectively purchased. Hoechst 33342, trihydrochloride, trihydrate (H3570) was from Invitrogen.

## Small interfering RNA (siRNA)-mediated gene silencing in HUVECs

Atg5 was silenced with smart pool siRNAs 5´-GGCAUUAUCCAAUUGGUUU-3´, 5´-GCAGAACCAUACUAUUUGC-3´, 5´-UGACAGAUUUGACCAGUUU-3´, 5´-ACAAAGAUGUGCUUCGAGA-3´ (#L-004374-00-0005, Dharmacon). FOXO1 was silenced with siRNA 5´AAGAGCTGCATCCATGGACAA-3´ (#SI04435144, Qiagen). Tie-2 was silenced with siRNA 5´-TCGGTGCTACTTAACAACTTA-3´ (#SI00604919, Qiagen). Allstar negative siRNA (Qiagen) was used as a negative control. HUVECs were transfected with 80 nM siRNAs for 48–72 h using Lipofectamine RNAiMAX (Invitrogen) according to manufacturer's instructions.

## Application of shear stress

HUVECs were plated on fibronectin-coated µ-Slide VI 0.4 (80606, ibidi GmbH), µ-Slide I Leuer 0.8 (80196, ibidi GmbH), µ-Slide I Leuer 0.4 (80176, ibidi GmbH) or µ-Slide I Leuer 0.2 (80166, ibidi GmbH) at a density of $1 \times 10^5$ cells/cm$^2$ and cultured for 48 h in EGM-2. The culture medium was replaced with EBM-2 medium containing 2% FBS and 1% penicillin/streptomycin 4 h before applying the shear stress. Shear stress was applied for 24 h (10902 Pump System, ibidi GmbH, Germany) at 37°C in the same humidified incubator.

## Immunofluorescence staining

HUVECs were fixed with 4% PFA in PBS for 15 minutes. Fixed cells were then permeabilized with 0.3% Triton X-100 in PBS for 5 minutes, blocked with 1% BSA in PBS for an hour at room temperature, followed by overnight incubation with primary antibodies at 4°C. Subsequently, cells were washed with PBS and incubated with secondary antibodies and Hoechst dye for an hour at room temperature. Images were obtained using a Zeiss LSM880 confocal microscope.

## Immunoblotting

HUVECs were lysed in buffer containing 50 mM Tris–HCl, pH 7.4, 150 mM NaCl, 2 mM EDTA, 0.5% Na-Deoxycholat, 0.1% SDS, 1% Triton X-100 and 1:50 cOmplete EDTA-free Proteinase Inhibitor Cocktail (Roche). Proteins were separated by SDS–PAGE and transferred to nitrocellulose membranes (Schleicher & Schuell) by wet blotting.

## Analysis of cell orientation and elongation

Confocal microscopic images were analyzed with ImageJ/Fiji (version 2.14.0/1.54f). To define cell shapes, the VE-cadherin channel was applied to "Subtract Background, rolling = 20" and the filter "Gaussian Blur at sigma = 3", followed by converting images to binary images with "Auto threshold with Huang method". To detect orientation and length of major and minor axis in each cell, we fitted ellipses to the cellular outline in binary images with "Analyze particle". We analyzed the orientation and cell elongation index of > 500 cells in at least 5 images per lane of an ibidi cell culture chamber.

### Analysis of FOXO1 nuclear levels

Confocal microscopic images were analyzed with ImageJ/Fiji. To prepare a nuclear mask, Hoechst 33342 images were applied to the filter "Gaussian Blur at sigma = 1.5", and converted to binary images with "Auto threshold with Huang method". Nuclear FOXO1 signals were extracted by applying the mask to images of FOXO1 using the "image Calculator".

### Analysis of LC3B puncta area per cell

Confocal microscopic images were analyzed with ImageJ/Fiji. To extract LC3B puncta, LC3B images were converted to binary images by thresholding the signal pixel intensity above 70. An area of LC3B puncta ($\mu m^2$) was measured, and LC3B puncta area per cell was calculated based on the number of nuclei in the image.

### Statistical analysis

Values are expressed as mean ± standard error of the mean (SEM). Differences between two groups were assessed by two-tailed Mann-Whitney U-test and among more than two groups by ANOVA followed by Tukey test for multiple comparisons (Prism 10.31). The statistical significance of differences is shown in the figure legends. P values < 0.05 were considered statistically significant.

## Results

### Tie-2 is essential for endothelial morphological responses to shear stress

We have recently shown that laminar shear stress induces polarized distribution and endocytosis of VE-PTP which leads to activation of Tie-2 at cell contacts which in turn protects against vascular leaks and atheroma formation [31]. To investigate whether Tie-2 is not only affected by shear stress, but even required for shear stress-induced changes in endothelial cell morphology, we tested whether interference with Tie-2 expression would affect endothelial cell alignment. To this end, we treated HUVEC either with Tie-2 siRNA or control siRNA and exposed them for 24 hrs to 15 dyn/cm² shear stress (silencing efficiency shown in S1 Fig). As shown in Fig 1A, cell alignment was clearly stimulated by flow in control cells, but not in Tie-2 siRNA treated cells. Measurement of cell orientation revealed that the frequency of cells aligned within 0–30° of the direction of flow was drastically reduced in Tie-2 siRNA treated cells (Fig 1B). Likewise, the elongation index strongly increased in flow-exposed control cells when compared to cells grown under static conditions, whereas this increase was much weaker in Tie-2 siRNA treated cells (Fig 1C). These results reveal that Tie-2 is essential for endothelial morphological responses to shear stress.

### Tie-2 controls endothelial morphology under shear stress through nuclear exclusion of FOXO1

We next addressed how Tie-2 regulates the morphological responses. We have previously found that Tie-2 is involved in nuclear exclusion of the transcription factor FOXO1 upon short term exposure of EC to shear stress [31]. In addition, angiopoietin-based activation of Tie-2 was shown to inhibit the transcriptional activity of FOXO1 [42,44,45]. Therefore, we tested whether Tie-2 may control flow-induced endothelial cell alignment via FOXO1. First, we assessed the effect of Tie-2 on FOXO1 subcellular localization in aligned and elongated HUVEC after 24 h exposure to 15 dyn/cm² shear stress. In cells transfected with control siRNA, shear stress reduced nuclear FOXO1 levels along with cell morphological changes, whereas Tie-2 silencing strongly blocked the reduction of nuclear FOXO1 (Figs 2A and B). These findings reveal that Tie-2 is needed for shear stress-dependent nuclear exclusion and for keeping FOXO1 continuously away from the nucleoplasm.

We then examined whether FOXO1 contributes to shear stress induced morphological responses of endothelial cells. Besides the question whether FOXO1 is at all relevant for the process, it was particularly interesting to know whether it is loss of function of FOXO1 in the nucleus or gain of function of FOXO1 in the cytoplasm which is involved in morphological responses to flow. To test this, we examined shear stress induced cell alignment and cell elongation for HUVEC that

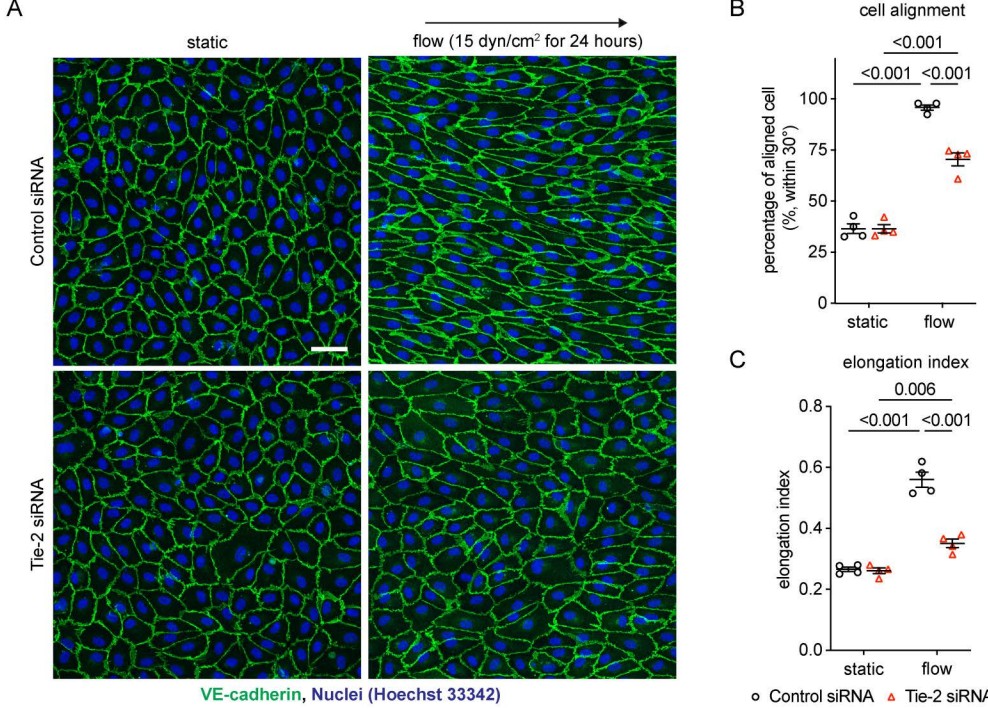

**Fig 1. Tie-2 is required for the morphological responses of endothelial cells triggered by shear stress.** (A–C) HUVECs transfected with control or Tie-2 siRNA were exposed to 15 dyn/cm$^2$ of shear stress for 24 hours. The resulting cells were stained for VE-cadherin (green) (A). Percentage of aligned cell to shear stress direction within 30° **(B)** and elongation index **(C)** were quantified using ImageJ/Fiji. Data presented as mean ± SEM. P values were calculated with two-way ANOVA followed by Tukey's multiple test. Scale bar: 50 μm.

were either treated with FOXO1 siRNA or control siRNA (silencing efficiency shown in S2 Fig). We found that silencing of FOXO1 indeed strongly inhibited cell alignment and cell elongation, as is illustrated in micrographs of cells stained for VE-cadherin and FOXO1 (Fig 3A) and documented by quantitative evaluation of the frequency of aligned cells and by determining the elongation index of these cells (Figs 3B-C). Thus, although the loss of FOXO1 in the nucleus accompanies endothelial cell alignment and elongation, FOXO1 is needed for these morphological changes in cell morphology. This strongly suggested that some function of FOXO1 in the cytoplasm is needed for cellular morphological responses to shear stress rather than FOXO1 transcriptional activity in the nuclei.

### Shear stress-induced autophagosome formation depends on Tie-2 and nuclear-excluded FOXO1 and is needed for cell alignment

We next examined how cytoplasmic FOXO1, after exclusion from the nucleus, influences the endothelial morphological responses to shear stress. It was previously shown that cells protect themselves against oxidative stress-induced damage by autophagy which is regulated by mechanisms that require cytosolic FOXO1 [38–40] Since autophagy was also described as being important for endothelial morphological responses to shear stress [46], we tested whether Tie-2 and FOXO1 would possibly be relevant for shear stress induced autophagy. To assess autophagosome formation, we conducted immunofluorescence staining of LC3B, a protein which is a well characterized factor relevant for autophagy [47]. To test our system, we first confirmed that the mTOR inhibitor Rapamycin increases the immunoreactivity for LC3B in our HUVEC cultures (S3 Fig). Next, we treated HUVEC either with control siRNA or Tie-2 or FOXO1 siRNA, followed by exposure to 15 dyn/cm$^2$ shear stress for 24 h and stained for LC3B. We found that shear stress exposure strongly increased

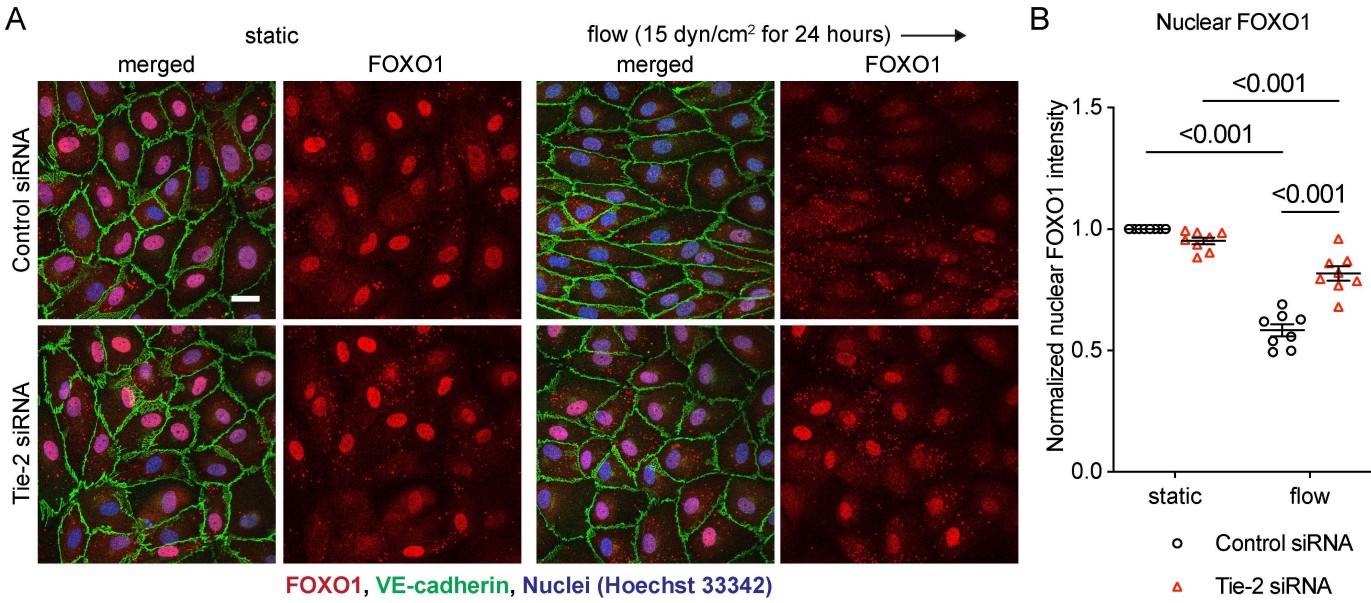

**Fig 2. Tie-2 is required for nuclear exclusion of FOXO1 induced by shear stress.** (A, B) HUVECs transfected with control or FOXO1 siRNA were exposed to 15 dyn/cm² of shear stress for 24 hours. The resulting cells were stained for FOXO1 (red) and VE-cadherin (green)**(A)**. Quantification of nuclear FOXO1 intensity with ImageJ/Fiji (B). Data presented as mean±SEM. P values were calculated with two-way ANOVA followed by Tukey's multiple test. Scale bar indicates 25 μm.

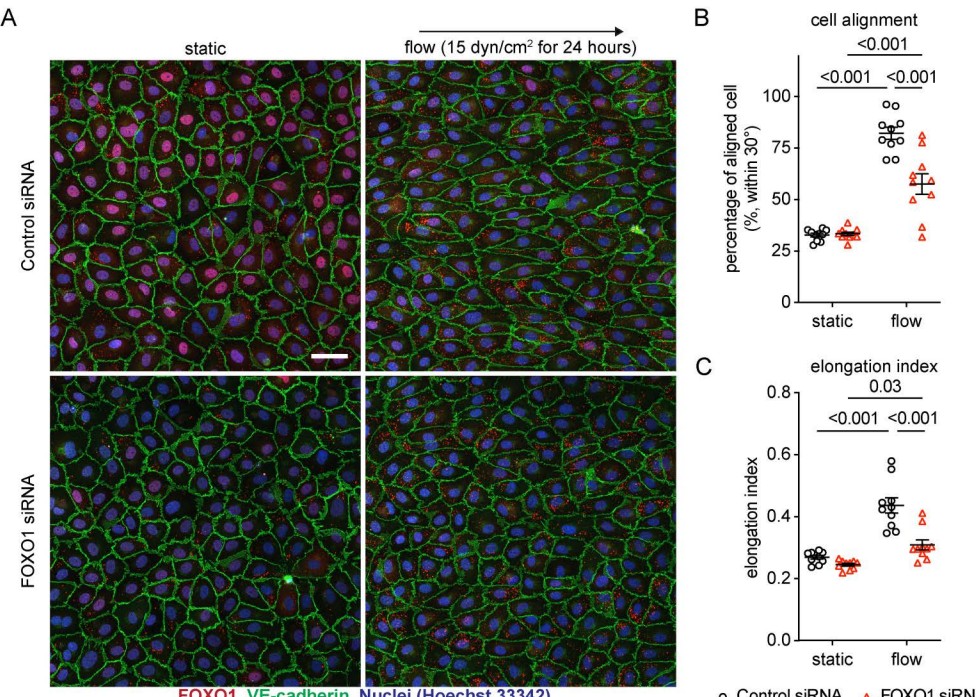

**Fig 3. FOXO1 is required for cell morphological responses induced by shear stress.** (A-D) HUVECs transfected with control or FOXO1 siRNA were exposed to 15 dyn/cm² shear stress for 24 hours. The resulting cells were stained for FOXO1 (red) and VE-cadherin (green) (A). Percentage of aligned cell to shear stress direction within 30° **(B)** and elongation index **(C)** were quantified with ImageJ/Fiji. Data presented as mean±SEM. P values were calculated with two-way ANOVA followed by Tukey's multiple test. Scale bar indicates 50 μm.

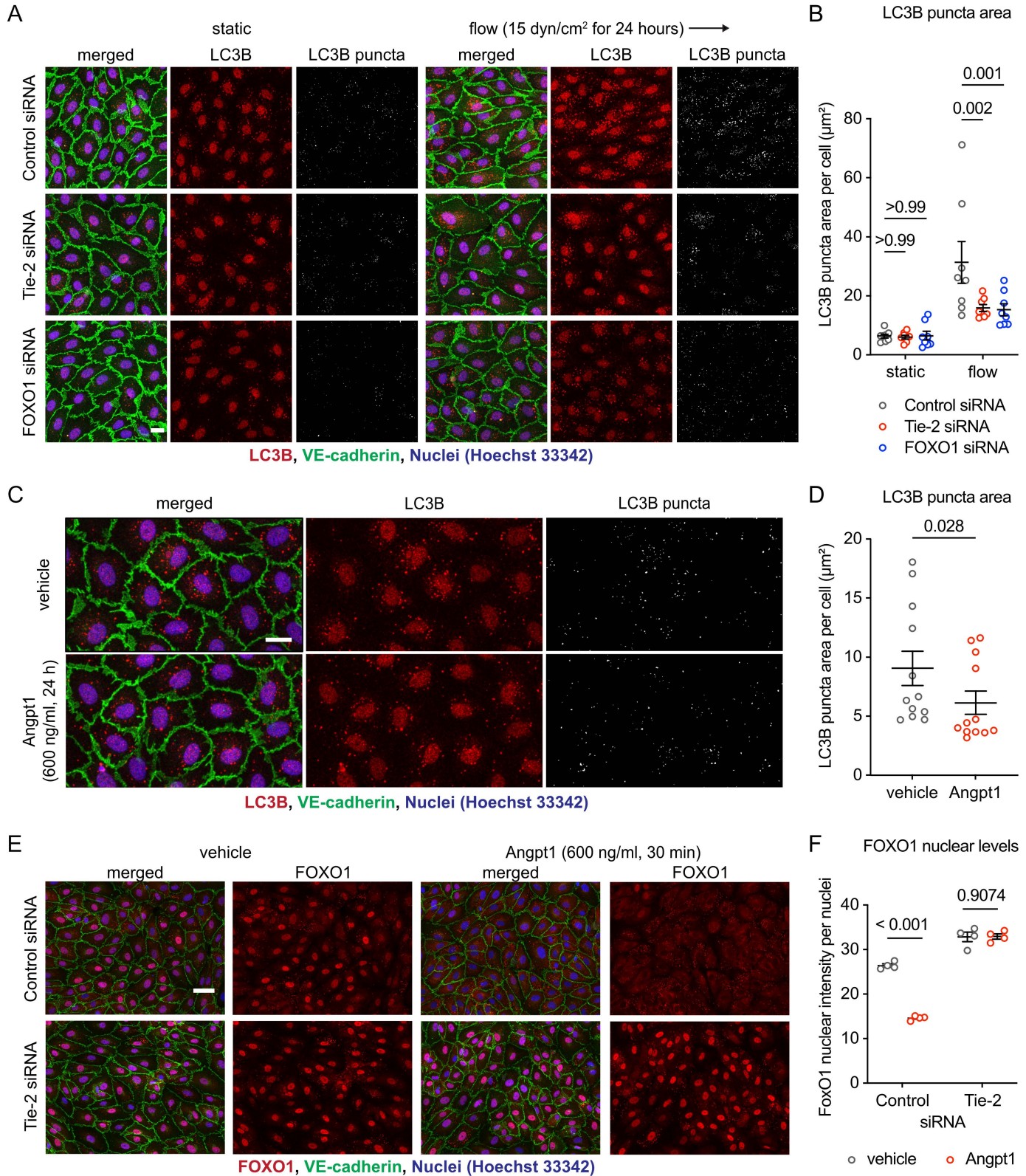

**Fig 4. Shear stress-stimulated Tie-2-FOXO1 signaling regulates autophagosome formation.** (A-B) HUVECs transfected with control, Tie-2 or FOXO1 siRNA were exposed to shear stress at 15 dyn/cm² for 24 hours. The resulting cells were stained for LC3B (red) and VE-cadherin (green) (A).

Areas of LC3B puncta in each cell were quantified with ImageJ/Fiji (B) (C–D) HUVECs were treated with 600 ng/ml Angpt1 for 24 hrs followed by staining for LC3B (red) and VE-cadherin (green) (C). Areas of LC3B puncta in each cell were quantified with ImageJ/Fiji (D). (E–F) HUVECs transfected with control or Tie-2 siRNA were treated with 600 ng/ml Angpt1 for 30 minutes, followed by staining for FOXO1 (red) and VE-cadherin (green) (E). Nuclear FOXO1 levels were quantified with ImageJ/Fiji (F). Data presented as mean ± SEM. P values were calculated with two-way ANOVA followed by Tukey's multiple test (**B and F**) or Mann–Whitney U test (D). Scale bars indicate 20 μm (A and C) or 50 μm (E).

the area of LC3B puncta in the cytoplasm of HUVEC treated with control siRNA, whereas silencing of Tie-2 and FOXO1 strongly reduced this increase (Fig 4A). Reduction of this increase was highly significant although not complete (Fig 4B). We then tested whether direct stimulation of Tie-2 by an agonist would be sufficient for the formation of autophagosomes. Interestingly, we found that Angpt1 treatment of HUVEC was not sufficient to induce LC3B puncta (Figs 4C and D). In contrast, Angpt1 stimulation was sufficient to trigger nuclear exclusion of FOXO1 (Figs 4E and F), which is consistent with a previous study [48]. Thus, Tie-2 stimulated FOXO1 nuclear exclusion is needed and strongly involved in shear stress-induced formation of autophagosomes, but is not sufficient as autophagy stimulus in the absence of shear stress.

In order to test whether autophagosome formation is indeed relevant for shear stress-induced cell alignment and elongation in our experimental system, we treated HUVEC with control siRNA and siRNA against Atg5, a key protein for the induction of autophagy. The efficiency of Atg5 siRNA treatment was tested by immunoblotting for Atg5 (Fig 5A) and by comparing the amounts of activated and inactivated LC3B as detected by immunoblotting (Fig 5B). We found that cell alignment was strongly reduced, as illustrated in micrographs of cells stained for VE-cadherin and cell nuclei (Fig 5C). Quantification revealed a significant reduction of cell alignment (Figs 5D), whereas cell elongation was not affected by Atg5 siRNA treatment (Fig 5E). These results of Fig. 5 demonstrate the importance of autophagy for cell alignment but not elongation in our HUVEC culture system. Collectively, our results suggest that Tie-2 is required for flow-induced cell alignment by stimulating FOXO1 nuclear exclusion, which in turn stimulates autophagosome formation (Fig 6).

## Discussion

This study investigated whether Tie-2 plays a role for shear stress-induced changes of endothelial cell morphology. We found that Tie-2 was indispensable for shear stress-induced cell alignment and cell elongation. Mechanistically, we could show that Tie-2 supports these morphological changes by stimulating FOXO1 nuclear exclusion. Upon translocation into the cytosol, FOXO1 stimulated autophagosome induction which in turn was required for shear stress-induced changes in cell morphology. Thus, our results establish a novel Tie-2-FOXO1-autophagy signaling axis which is required for shear stress-induced cell alignment and elongation. Interestingly, despite the need of this axis for cell alignment, we found that neither Tie-2 stimulation nor cytosolic FOXO1 were alone sufficient to activate autophagy and cell alignment. Instead, they are necessary signaling steps that act in cooperation with other shear stress-induced signals which are additionally needed to trigger autophagy and cell alignment.

Although it was hitherto unknown that Tie-2 is required for flow-induced cell alignment, it was shown before that shear stress affects transcription of Tie-2 and its ligands [26–29], and in the case of Tie-2 this was shown to affect cell cycle progression in synergy with TGF-β/BMP signaling [26]. In addition, laminar fluid shear stress was shown to stimulate Tie-2 activity [30] and this was shown to be driven by polarized distribution and sequestration of the phosphatase VE-PTP [31]. This activation of Tie-2 enhanced vascular integrity and protected against vascular leaks and the formation of atherosclerotic plaques [31,32]. The need of Tie-2 for cell alignment is a novel function of Tie-2 in response to shear stress. Since cell alignment in response to laminar shear stress helps to reduce the shear stress sensed by a cell, this is a novel way how Tie-2 protects against inflammatory cell activation.

FOXO1 is a transcription factor with numerous activities during vascular development [33–37]. While it was shown before that Tie-2 stimulation by Angpt1 induces nuclear exclusion of FOXO1, this effect was only analyzed so far with respect to its consequences on transcriptional regulation [42,44]. Furthermore, laminar shear stress was shown to

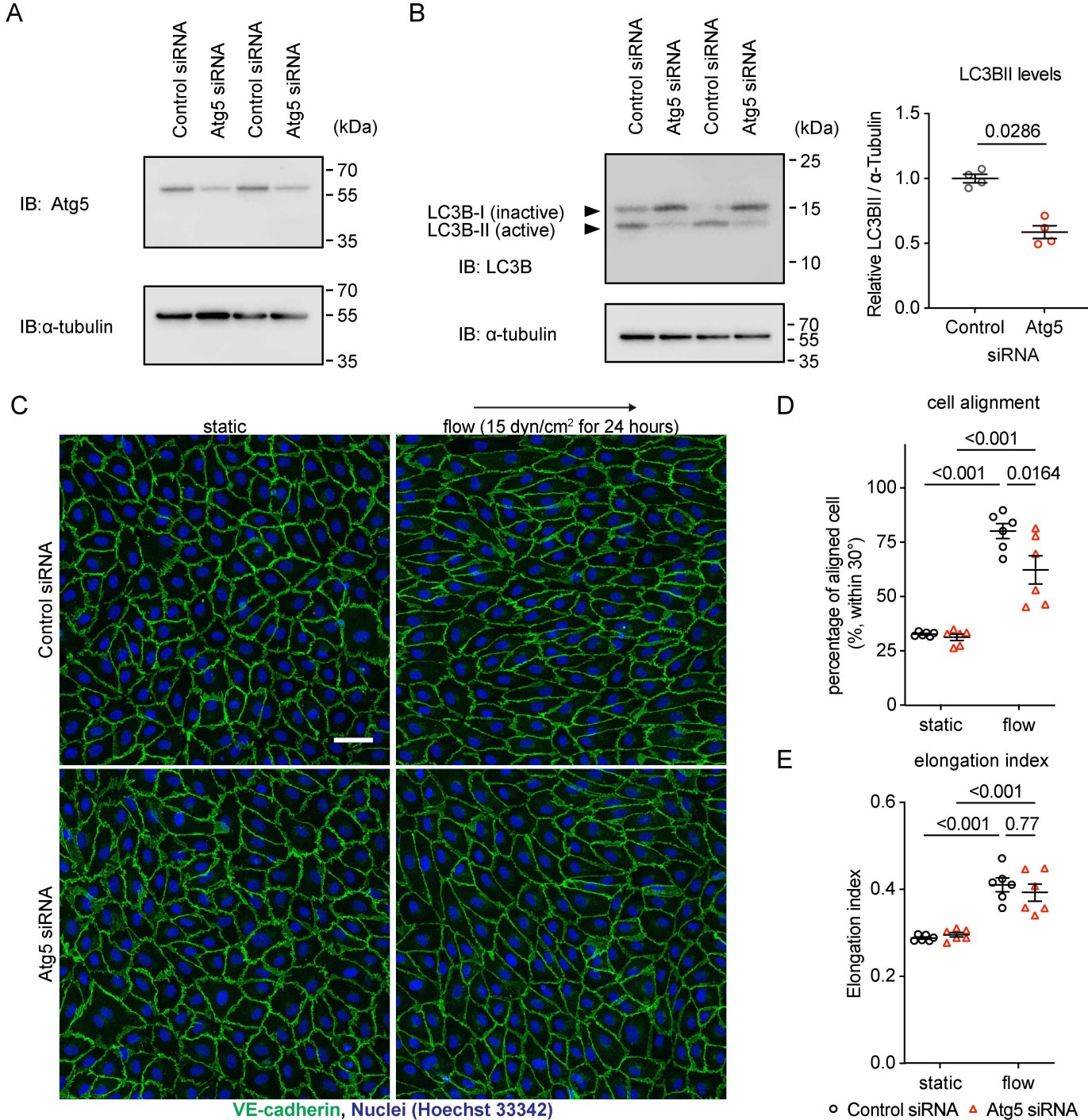

**Fig 5. Autophagy is required for cell alignment under laminar shear stress.** (A) To test siRNA efficiency, lysates of control or Atg5 siRNA treated HUVEC were immunoblotted for Atg5 and α-tubulin as a loading control 48 hours later. (B) To test the effect of Atg5 silencing on LC3B activity, lysates of control or Atg5 siRNA-treated HUVEC were immunoblotted for LC3B and α-tubulin as a loading control 48 hours later. The inactive and active forms of LC3B were distinguished by their electrophoretic mobility. Quantification is shown on the right. (C-E) HUVECs transfected with control or Atg5 siRNA were exposed to 15 dyn/cm² shear stress for 24 hours followed by staining for VE-cadherin (green) (C). Percentage of aligned cell to shear stress direction within 30° **(D)** and elongation index **(E)** were quantified with ImageJ/Fiji. Data are presented as mean±SEM. P values were calculated with Mann–Whitney U test (**B**) and two-way ANOVA followed by Tukey's multiple test (D and E). Scale bars indicate 50 μm.

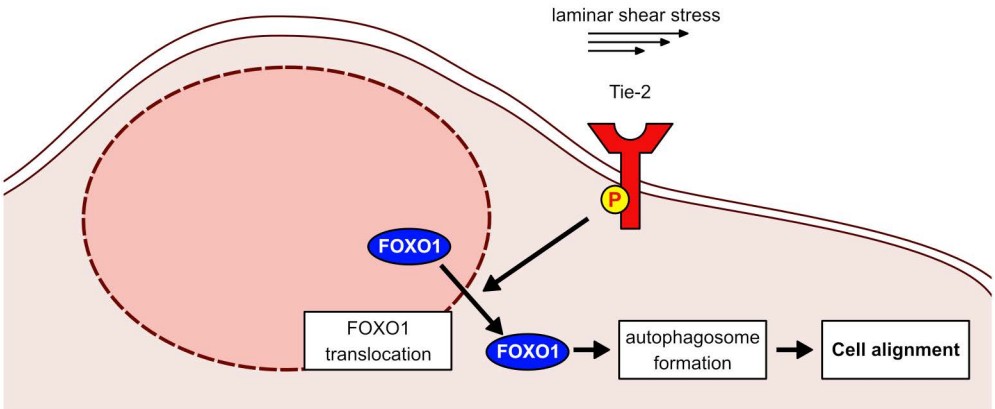

**Fig 6. Proposed mechanism by which Tie-2 regulates the morphological responses of endothelial cells to shear stress via FOXO1.** Laminar shear stress stimulates the activation of Tie-2, which leads to the translocation of FOXO1 from the nucleus to the cytosol where it triggers autophagosome formation which supports cell alignment.

trigger nuclear exclusion of FOXO1, yet it was not known that this is mediated by Tie-2 and again it was only analyzed for its consequences on gene transcription [29,32]. In contrast, we show here that it is Tie-2 which is needed for shear stress-induced FOXO1 nuclear exclusion. In addition, we show that FOXO1 is required for cell alignment, and that this supportive function was blocked upon silencing its expression. This clearly suggests that it is the cytosolic FOXO1 which is needed for this function and not FOXO1 in the nucleus.

Cytosolic FOXO1 was shown in various cell types to be required for the induction of autophagy by various processes different from shear stress induced signaling [38–41]. Interestingly, in carcinoma cells it was sufficient to express a cytosolic variant of FOXO1 in order to induce autophagosome formation (LC3B puncta formation). In contrast, we found that Tie-2 stimulation with Angpt1 in endothelial cells, although sufficient to trigger FOXO1 nuclear exclusion, was not sufficient to stimulate autophagosome formation. Thus, in primary endothelial cells cytosolic FOXO1 seems to be insufficient to stimulate autophagocytosis. Additional signals, provided by shear stress exposure of the cells seem to be required besides Tie-2 and FOXO1, which are required, but not sufficient.

Endothelial autophagy was shown before to be involved in shear stress induced cell alignment [46]. Investigating mechanosensing mechanisms at the cell surface for a possible role in autophagy induction revealed that neither the well documented signaling unit PECAM-1-VE-cadherin-VEGFR2 [7] nor the primary cilium were required for shear stress-induced autophagy [46]. Thus, it is remarkable that the Tie-2-FOXO1 axis which we describe here is essential for this mechanosensing signaling process. This highlights the importance of the multiplicity of shear sensing and signaling mechanisms which can complement each other according to the physiological needs.

In summary, our study shows that Tie-2-FOXO1-autophagy constitutes a critical signaling response which supports cell alignment in the direction of fluid shear stress. This extends our understanding of how Tie-2 contributes to vascular protection under laminar shear stress.

## Supporting information

**S1 Fig. siRNA-mediated Tie-2 knock-down efficiency.** Immunoblotting for Tie-2 and α-tubulin as a loading control of HUVECs transfected with control or Tie-2 siRNA for 48 hours. Data are representative for three similar experiments. (PDF)

**S2 Fig. siRNA-mediated FOXO1 knock-down efficiency.** Immunoblotting for FOXO1 and α-tubulin as a loading control of HUVECs transfected with control or FOXO1 siRNA for 48 hours. Data are representative for three similar experiments. (PDF)

**S3 Fig. LC3B antibody validation LC3B immunofluorescence staining of HUVECs treated with rapamycin at 1 μM for 24 hours.** Scale bar indicates 50 μm. Data are representative for three similar experiments. (PDF)

**S1 Raw Data. Source data underlying figure panels in the paper.** (XLSX)

## Acknowledgments

We thank Prof. Hans-J. Schnittler (Münster University, Germany) for providing HUVECs.

## Author contributions

**Conceptualization:** Keisuke Shirakura.

**Data curation:** Mana Ghanbarpour Houshangi, Keisuke Shirakura.

**Formal analysis:** Mana Ghanbarpour Houshangi, Keisuke Shirakura.

**Funding acquisition:** Dietmar Vestweber.

**Investigation:** Mana Ghanbarpour Houshangi, Keisuke Shirakura.

**Methodology:** Keisuke Shirakura.

**Project administration:** Keisuke Shirakura, Dietmar Vestweber.

**Supervision:** Keisuke Shirakura, Dietmar Vestweber.

**Validation:** Mana Ghanbarpour Houshangi, Keisuke Shirakura.

**Visualization:** Mana Ghanbarpour Houshangi, Keisuke Shirakura.

**Writing – original draft:** Mana Ghanbarpour Houshangi, Keisuke Shirakura.

**Writing – review & editing:** Keisuke Shirakura, Dietmar Vestweber.

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
