## [Editor Report · Decision Letter 0]

30 Oct 2024

PONE-D-24-43059Tie-2 regulates endothelial morphological responses to shear stress by FoxO1-triggered autophagyPLOS ONE

Dear Dr. Shirakura,

Thank you for submitting your manuscript to PLOS ONE. After careful consideration, we feel that it has merit but does not fully meet PLOS ONE’s publication criteria as it currently stands. Therefore, we invite you to submit a revised version of the manuscript that addresses the points raised during the review process.

We look forward to receiving your revised manuscript.

Kind regards,

Saiedeh Razi-Soofiyani

Academic Editor

PLOS ONE

Journal Requirements: When submitting your revision, we need you to address these additional requirements. 1. Please ensure that your manuscript meets PLOS ONE's style requirements, including those for file naming. The PLOS ONE style templates can be found at https://journals.plos.org/plosone/s/file?id=wjVg/PLOSOne_formatting_sample_main_body.pdf and https://journals.plos.org/plosone/s/file?id=ba62/PLOSOne_formatting_sample_title_authors_affiliations.pdf 2. PLOS ONE now requires that authors provide the original uncropped and unadjusted images underlying all blot or gel results reported in a submission’s figures or Supporting Information files. This policy and the journal’s other requirements for blot/gel reporting and figure preparation are described in detail at https://journals.plos.org/plosone/s/figures#loc-blot-and-gel-reporting-requirements and https://journals.plos.org/plosone/s/figures#loc-preparing-figures-from-image-files. When you submit your revised manuscript, please ensure that your figures adhere fully to these guidelines and provide the original underlying images for all blot or gel data reported in your submission. See the following link for instructions on providing the original image data: https://journals.plos.org/plosone/s/figures#loc-original-images-for-blots-and-gels.   In your cover letter, please note whether your blot/gel image data are in Supporting Information or posted at a public data repository, provide the repository URL if relevant, and provide specific details as to which raw blot/gel images, if any, are not available. Email us at plosone@plos.org if you have any questions. 3. Please ensure that you refer to Figure 6 in your text as, if accepted, production will need this reference to link the reader to the figure. 4. Please include captions for your Supporting Information files at the end of your manuscript, and update any in-text citations to match accordingly. Please see our Supporting Information guidelines for more information: http://journals.plos.org/plosone/s/supporting-information. 

**Additional Editor Comments:**

The subject of the paper is interesting, but needs revision to be acceptable.

1. Abstract: please rewrite the abstract section based on following format: Introduction, Methods, Results and conclusion. You should not your previous findings in this part.

2. Please use Mesh terms as keywords and put them based on alphabetic order.

3. Introduction section is boring, please summarize this part in 1 page.

4. In Methods section, please indicate the complete information of the software which used.

5. Try to put . at the end of sentences.

6. Compare your results with more slimier studies, what is difference or similarity?

7. Use updated references, Please replace out of date references with updated ones.

8. What is control protein for western blotting?

---

## [Author Response · Author response to Decision Letter 1]

6 Dec 2024

Response to reviewers

We thank the editor and referees for their comments to our manuscript. We have addressed the comments raised by the editor as given below.

Comment 1. Abstract: please rewrite the abstract section based on following format: Introduction, Methods, Results and conclusion. You should not your previous findings in this part.

Response: We modified the abstract according to your comments.

Comment 2. Please use Mesh terms as keywords and put them based on alphabetic order.

Response: We now added MeSH terms as keywords:

Blood vessel (A07.015), Autophagy (G04.011), Cell shape (G04. 320), Signal Transduction (G04.835) and Mechanotransduction, Cellular (G04.835.580).

Comment 3. Introduction section is boring, please summarize this part in 1 page.

Response: We revised the introduction section accondingly.

Comment 4. In Methods section, please indicate the complete information of the software which used.

Response: We added the information about the versions of all software we used in this study.

Comment 5. Try to put . at the end of sentences.

Response: We did this now.

Comment 6. Compare your results with more slimier studies, what is difference or similarity?

Response: We have now re-written parts of the discussion and descibed in more detail what is novel and how this fits to published results.

Comment 7. Use updated references, Please replace out of date references with updated ones.

Response: We revised the references accoringg to the style of PLOS ONE.

Comment 8. What is control protein for western blotting?

Response: In this study, we used alpha-tubulin as a control proten for western blotting. To clarify this, we revised the figure legends.

---

## [Decision Letter · Decision Letter 1]

13 Feb 2025

PONE-D-24-43059R1Tie-2 regulates endothelial morphological responses to shear stress by FoxO1-triggered autophagyPLOS ONE

Dear Dr. Shirakura,

Thank you for submitting your manuscript to PLOS ONE. After careful consideration, we feel that it has merit but does not fully meet PLOS ONE’s publication criteria as it currently stands. Therefore, we invite you to submit a revised version of the manuscript that addresses the points raised during the review process. Please submit your revised manuscript by Mar 30 2025 11:59PM. If you will need more time than this to complete your revisions, please reply to this message or contact the journal office at plosone@plos.org. Please include the following items when submitting your revised manuscript:

We look forward to receiving your revised manuscript.

Kind regards,

Saiedeh Razi-Soofiyani

Academic Editor

PLOS ONE

Journal Requirements:

Additional Editor Comments:

Please revise the paper based on reviewers comments.

Reviewers' comments:

Reviewer's Responses to Questions

**Comments to the Author**

1. If the authors have adequately addressed your comments raised in a previous round of review and you feel that this manuscript is now acceptable for publication, you may indicate that here to bypass the “Comments to the Author” section, enter your conflict of interest statement in the “Confidential to Editor” section, and submit your "Accept" recommendation.

Reviewer #1: (No Response)

Reviewer #2: All comments have been addressed

2. Is the manuscript technically sound, and do the data support the conclusions?

Reviewer #1: Partly

Reviewer #2: Yes

3. Has the statistical analysis been performed appropriately and rigorously?

Reviewer #1: I Don't Know

Reviewer #2: Yes

4. Have the authors made all data underlying the findings in their manuscript fully available?

Reviewer #1: Yes

Reviewer #2: Yes

5. Is the manuscript presented in an intelligible fashion and written in standard English?

Reviewer #1: Yes

Reviewer #2: Yes

6. Review Comments to the Author

Reviewer #1: This study investigated whether and how Tie2-FoxO1 axis is involved in flow-induced morphological changes in EC. There are several major concerns about the methodology used and the validity of conclusions. Moreover, the quality of some data need to be improved.

Major issues

Figures 1B, 3B, 5B lack statistical results; we cannot identify whether the effects are significant or not.

Similar to the data in Fig S1 and S2, the gene silencing efficacy by Tie2 siRNA should be presented.

Please clarify whether the data in FigS1 and S2 were from a single experiment only? If this is the case, these results should be clearly labeled as preliminary.

Data in Fig 4A to 4D: These fluorescent images of LC3B are not representative; it is difficult to identify the differences between groups, despite the numerical data showing statistical significances. High resolution images which highlight the differences in more detail are required to validate the numerical data. Moreover, autophagy is characterized by formation of LC3 puncta; it is not clear what “LC3 area” stands for, and how it is measured. These methodological details are critical and need to be added.

Figure 5A to 5C: it is obvious that Atg5 silencing had little effects on flow/Tie2-induced EC morphogenesis responses, indicating that autophagy induction is not mediating the actions. These data appear to be against the conclusion proposed by the authors.

Data analysis: A number of tests and post hoc detection methods were mentioned in the methods section. However, as indicated in the figure legends, only two methods (two-way ANOVA followed by Tukey’s multiple test and Mann–Whitney U test) were used. Please clarify.

Minor points

Page 9 line 178-181 (in the word document): Besides the question, whether FoxO1 is at all involved in the process, it was particularly interesting to know, whether it is the loss of the transcriptional activity of FoxO1 in the nucleus or whether it is a function of FoxO1 in the cytoplasm which involved in morphogenesis responses to flow. This sentence is hard to follow, please rephrase.

Page 9-10 line 188-190: This strongly suggests that it is not the inhibition of FoxO1 transcriptional activity in the is some function of FoxO1 in the cytoplasm which is needed for cellular morphological responses to shear stress. This sentence is hard to follow, please rephrase.

Figure 5: Results of validation of Atg5 silencing on autophagosome are to be presented first, well before its main actions on EC morphology.

Reviewer #2: This study identifies a novel Tie-2-FoxO1-autophagy pathway critical for endothelial alignment under shear stress. While the data are compelling, addressing the above points will enhance mechanistic clarity, validate key conclusions, and broaden the impact of the work. However, the paper still needs some improvements before acceptance for publication.

My detailed comments are as follows:

1. I don’t think “FoxO1” in the title is the standard name, FOXO1 is more commonly used in the literature, please check that.

2. Could the authors provide any evidence to expose HUVECs for 24 hrs to 15 dyn/cm2 shear?

3. Why does Atg5 silencing only partially inhibit alignment (e.g., test additional autophagy genes like ULK1 or Beclin-1)?

4. Clarify the rationale for Angpt1 concentration in 600 ng/ml, please cite previous studies or pre-experimental optimization.

5. This revised version still has some typos, such as “mechansims” should be “mechanisms”, “microgrpahs” should be “micrographs”.

7. PLOS authors have the option to publish the peer review history of their article (what does this mean? ). If published, this will include your full peer review and any attached files.

**Do you want your identity to be public for this peer review?** For information about this choice, including consent withdrawal, please see our Privacy Policy .

Reviewer #1: **Yes: ** Fan JIANG

Reviewer #2: No

---

## [Author Response · Author response to Decision Letter 2]

17 Mar 2025

Point by point response to reviewer’s comments

We thank the reviewers for their positive and very constructive comments that we have addressed below as follows:

Reviewer #1

This study investigated whether and how Tie2-FoxO1 axis is involved in flow-induced morphological changes in EC. There are several major concerns about the methodology used and the validity of conclusions. Moreover, the quality of some data need to be improved.

Major issues

Figures 1B, 3B, 5B lack statistical results; we cannot identify whether the effects are significant or not.

The way we had presented our cell alignment data made it difficult to include statistical significance into the graph. We have now chosen a different way to present these data by determining the percentage of cells that align within 30° of flow direction. Statistical significance is clearly seen in all results shown in the new figures 1B, 3B, and 5D. These results demonstrate the prominent effects of Tie2, FOXO1, and Atg5 in controlling cell alignment.

Similar to the data in Fig S1 and S2, the gene silencing efficacy by Tie2 siRNA should be presented.

We have now documented the gene silencing efficacy of Tie-2 siRNA by immunoblotting in the new Figure S1.

Please clarify whether the data in FigS1 and S2 were from a single experiment only? If this is the case, these results should be clearly labeled as preliminary.

In each of the two supplemental figures data are representative of three similar, independent experiments. To clarify that, we modified the figure legends.

Data in Fig 4A to 4D: These fluorescent images of LC3B are not representative; it is difficult to identify the differences between groups, despite the numerical data showing statistical significances. High resolution images which highlight the differences in more detail are required to validate the numerical data. Moreover, autophagy is characterized by formation of LC3 puncta; it is not clear what “LC3 area” stands for, and how it is measured. These methodological details are critical and need to be added.

We now show higher magnifications for the images in Fig. 4A and C, which makes it indeed much easier to interpret them.

We also explain more clearly now what we mean by “LC3 area”. In this study, we have measured the LC3B puncta area per cell as the reviewer mentioned. We applied a specific sensitivity threshold to the LC3B channel and extracted pixels with high signal intensity (Figure 4A and C; LC3B puncta column). We measured the area of these pixels and evaluated autophagosome formation in this way. The same method was used in recent studies conducted by the groups of Rubinsztein and Sparrer (Sci Rep. 2019, PMID: 31300716; Sci Rep. 2020, PMID: 32699244; Nat Chem Biol. 2021 PMID: 33510452). To explain and document this method in more detail, we added new binary images based on the LC3B channel in Figure 4A and 4C, and we also explained the detailed method in the Materials and Methods section.

Figure 5A to 5C: It is obvious that Atg5 silencing had little effect on flow/Tie2-induced EC morphogenesis responses, indicating that autophagy induction does not mediate the actions. These data appear to be against the conclusion proposed by the authors.

We very much appreciate that the reviewer pointed this out. We mixed up the two images for ctrl siRNA and ATG5 siRNA of cells under flow in Fig. 5C (previous Fig. 5A). We apologize for this oversight.

According to Figure 5C, D and E, Atg5 siRNA significantly reduced cell alignment. This is consistent with the data of Tie-2 and FOXO1. In contrast to cell alignment, Atg5 siRNA did not affect cell elongation (Figure 5E) although both, Tie2 and FOXO1 siRNA suppressed cell elongation significantly. Based on these data, we conluded autophagy is the responsible mechanism by which Tie2-FOXO1 controls cell alignment.

Previous studies have reported FoxO1 interacts with Autophagy Related 7 (Atg7), leading to autophagosome formation through Atg5 (Nat Cell Biol. 2010, PMID: 20543840; Nat Commun 2016, PMID: 27010363; Cell Cycle 2022, PMID: 36510368). Endothelial Atg7 has been also reported to regulate fibronectin expression in an autophagy-independent manner (J Cell Biol 2023, PMID: 36995368). Fibonectin has been known to affect endothelial shear response via integrins (EMBO J. 2001, PMID: 11532928). Thus, Tie-2 and FoxO1 may regulate cell elongation through this aspect, but this would be a future consideration.

To clarify our finding and conclusion in this study, we modified the text at line 233-239 on page11.

Data analysis: A number of tests and post hoc detection methods were mentioned in the methods section. However, as indicated in the figure legends, only two methods (two-way ANOVA followed by Tukey’s multiple test and Mann–Whitney U test) were used. Please clarify.

We appreciate that the reviewer pointed out this error. We corrected the error in the Materials and Methods section. Indeed, only two methods were used.

Minor points

Page 9 line 178-181 (in the word document): Besides the question, whether FoxO1 is at all involved in the process, it was particularly interesting to know, whether it is the loss of the transcriptional activity of FoxO1 in the nucleus or whether it is a function of FoxO1 in the cytoplasm which involved in morphogenesis responses to flow. This sentence is hard to follow, please rephrase.

We rephrased the sentence to the following: “Besides the question whether FOXO1 is at all relevant for the process, it was particularly interesting to know whether it is loss of function of FOXO1 in the nucleus or gain of function of FOXO1 in the cytoplasm which is involved in morphological responses to flow.”

Page 9-10 line 188-190: This strongly suggests that it is not the inhibition of FoxO1 transcriptional activity in the is some function of FoxO1 in the cytoplasm which is needed for cellular morphological responses to shear stress. This sentence is hard to follow, please rephrase.

We rephrased the sentence to the following: “This strongly suggested that some function of FOXO1 in the cytoplasm is needed for cellular morphological responses to shear stress rather than FOXO1 transcriptional activity in the nucleus.”

Figure 5: Results of validation of Atg5 silencing on autophagosome are to be presented first, well before its main actions on EC morphology.

We changed the order of items in Figure 5 and modified the text accordingly.

Reviewer #2

This study identifies a novel Tie-2-FoxO1-autophagy pathway critical for endothelial alignment under shear stress. While the data are compelling, addressing the above points will enhance mechanistic clarity, validate key conclusions, and broaden the impact of the work. However, the paper still needs some improvements before acceptance for publication.

My detailed comments are as follows:

1. I don’t think “FoxO1” in the title is the standard name, FOXO1 is more commonly used in the literature, please check that.

We agree with the reviewer and are now using the standard name FOXO1 throughout the text of the manuscript.

2. Could the authors provide any evidence to expose HUVECs for 24 hrs to 15 dyn/cm2 shear?

According to the literature, endothelium in venules is usually exposed to a physiological shear stress of 10–20 dyn/cm2 of shear stress (Hellenic J Cardiol 2005 PMID: 15807389). In line with this, 10–20 dyn/cm2 of shear stress induce cell alignment and elongation in HUVECs (eLife 2015 PMID: 25643397). Based on this, we employed 15 dyn/cm2 of shear stress to evaluate the effects of Tie-2 and FOXO1 in HUVECs.

3. Why does Atg5 silencing only partially inhibit alignment (e.g., test additional autophagy genes like ULK1 or Beclin-1)?

In response to Reviewer 1, we have now chosen a different way to present the quantitative analysis of cell alignment data. This presentation now reports the percentage of cells that align within 30° of flow direction. This analysis shows that Atg5 silencing inhibits cell alignment significantly by 22.3%. While this is indeed a partial inhibitory effect, it is in the same range as the effects we have seen for Tie2 silencing (26.5% reduction) or FOXO1 silencing (30 % reduction). The partial effect of Atg5 silencing on cell alignment is not due to a partial effect on autophagy since we show in Figure 5B (previous 5E) that autophagosome formation was very effectively blocked.

Statistical significance is clearly seen in all results shown in the new figures 1B, 3B and 5D. Based on the revised data, we modified the text at line 237 on page 11.

4. Clarify the rationale for Angpt1 concentration in 600 ng/ml, please cite previous studies or pre-experimental optimization.

In order to activate Tie2 strongly, we treated HUVECs with Angpt1 at 600 ng/ml according to a previous study (Proc Natl Acad Sci U S A 2018 PMID: 29358379). We cite this paper now on page 11 (line 227).

5. This revised version still has some typos, such as “mechansims” should be “mechanisms”, “microgrpahs” should be “micrographs”.

We corrected the errors in the text accordingly.

---

## [Editor Report · Decision Letter 2]

31 Mar 2025

Tie-2 regulates endothelial morphological responses to shear stress by FOXO1-triggered autophagy

PONE-D-24-43059R2

Dear Dr. Keisuke Shirakura,

We’re pleased to inform you that your manuscript has been judged scientifically suitable for publication and will be formally accepted for publication once it meets all outstanding technical requirements.

Kind regards,

Saiedeh Razi-Soofiyani

Academic Editor

PLOS ONE

Additional Editor Comments (optional):

The paper was revised precisely based on reviewers comments. Congratulation.
---

## [Editor Report · Acceptance letter]

PONE-D-24-43059R2

PLOS ONE

Dear Dr. Shirakura,

I'm pleased to inform you that your manuscript has been deemed suitable for publication in PLOS ONE. Congratulations! Your manuscript is now being handed over to our production team.

Kind regards,

on behalf of

Dr. Saiedeh Razi-Soofiyani

Academic Editor

PLOS ONE